# A Simple Expression for the Screening of Excitonic Couplings between Chlorophylls as Inferred for Photosystem I Trimers

**DOI:** 10.3390/ijms25169006

**Published:** 2024-08-19

**Authors:** Matthias Eder, Thomas Renger

**Affiliations:** Institute for Theoretical Physics, Johannes Kepler University Linz, Altenberger Str. 69, 4040 Linz, Austria; m.eder1@outlook.de

**Keywords:** excitonic couplings, dielectric screening, chlorophylls, light-harvesting antenna

## Abstract

The Coulomb coupling between transition densities of the pigments in photosynthetic pigment-protein complexes, termed excitonic coupling, is a key factor for the description of optical spectra and energy transfer. A challenging question is the quantification of the screening of the excitonic coupling by the optical polarizability of the environment. We use the equivalence between the sophisticated quantum chemical polarizable continuum (PCM) model and the simple electrostatic Poisson-TrEsp approach to analyze the distance and orientation dependence of the dielectric screening between chlorophylls in photosystem I trimers. On the basis of these calculations we find that the vacuum couplings $V_{mn}^{\left(0\right)}$ and the couplings in the dielectric medium $V_{mn}=f_{mn}V_{mn}^{\left(0\right)}$ are related by the empirical screening factor $f_{mn}=0.60+39.6\;\theta (|\kappa_{mn}\left|-1.17\right)\;\exp\left(-0.56\;R_{mn}/Å\right)$, where $\kappa_{mn}$ is the usual orientational factor of the dipole-dipole coupling between the pigments, $R_{mn}$ is the center-to-center distance, and the Heaviside-function $\theta (|\kappa_{mn}|-1.17)$ ensures that the exponential distance dependence only contributes for in-line type dipole geometries. We are confident that the present expression can be applied also to other pigment-protein complexes with chlorophyll or related pigments of similar shape. The variance between the Poisson-TrEsp and the approximate coupling values is found to decrease by a factor of 8 and 3–4 using the present expression, instead of an exponential distance dependent or constant screening factor, respectively, assumed previously in the literature.

## 1. Introduction

Photosynthesis provides the basis of life on earth and a play ground for developing theories and spectroscopy that are revealing the structure-function relationships of the photosynthetic machinery. Due to the stunning success of cryo-electron microscopy [1,2], we know the structural details of larger and larger units of the photosynthetic light-harvesting apparatus [3,4,5,6]. On one hand, these supercomplexes offer the opportunity to investigate long-distance energy transfer and, if a reaction center is present, trapping of excitation energy by primary electron transfer [7,8,9,10]. On the other hand, theories need to be developed that bridge the microscopic and the macroscopic world [9,11]. An important step towards such multi-scale theories concerns the development of approximate theories of excitation energy transfer and optical spectra and of approximate parameterization tools of the underlying Hamiltonians to treat large molecular system. These tools, ideally should be tested against more exact approaches on smaller systems, where the size of the reference system depends on the actual problem. As will be shown below, photosystem I (PSI) is of sufficient size (Figure 1) for the present problem.

In the last two decades, the theory of excitation energy transfer and optical spectra of photosynthetic light-harvesting and reaction center complexes has been developed such that in principle we are able now to treat the excitonic (pigment-pigment) coupling and the coupling of electronic transitions to inter- and intramolecular vibrational degrees of freedom on an equal footing [14,15,16,17,18,19,20,21]. In parallel, the structure-based parameterization of the Frenkel exciton Hamiltonian, used to describe these systems, has been advanced [22,23,24,25,26,27,28,29,30].

An important class of parameters concerns the excitonic couplings between pigments [22,27,31,32,33,34,35,36,37,38,39,40,41,42,43]. These couplings refer to the Coulomb coupling between transition densities, which can be obtained with quantum chemical methods. The first moment of the transition density is the transition dipole moment. For large interpigment distances, the excitonic coupling is given as the Coulomb coupling between transition dipole moments. For shorter distances, one has to go beyond this approximation and evaluate the Coulomb integral numerically, as in the transition density cube method [32], or in terms of atomic transition charges, as in the transition charges from electrostatic potential (TrEsp) method [35]. For very short distances, wavefunction overlap between pigments can give rise to additional contributions to the excitonic coupling, often caused by the coupling between local excited and charge transfer states [44,45,46,47]. In the present work, we will not deal with these short-range contributions but concentrate on long-range Coulomb effects, which dominate most of the excitonic couplings in pigment-protein complexes. There are a number of subtleties that need to be taken into account for an accurate description of the long-range excitonic couplings: Quantum chemical transition densities are qualitatively correct but need some rescaling to reproduce experimental transition dipole moments [48]. The value of the excitonic coupling, used in a calculation of optical spectra, depends on the theory of the optical spectra. A non-perturbative theory including high-frequency intramolecular modes needs to take into account the full transition density [20], whereas a perturbative theory without high-frequency modes just needs to consider the transition density of the 0-0 transition [49]. The influence of the polarizable environment on the excitonic couplings is threefold. First, the mutual polarization between the transition density of the pigments and the environment gives rise to reaction field effects that enhance the transition density [34,36,49]. Second, the pigment-induced polarization of the environment interacts with another pigment giving rise to screening effects [34,36,49]. Third, the polarization of the environment by the external field, used to measure the dipole strength of a pigment, gives rise to local field effects that influence the estimate of the dipole strength from experimental absorption data [49].

The main topic of the present work concerns the dependence of the dielectric screening on the mutual orientation of the pigments and their intermolecular distance. Investigations on selected dimers of photosystem II reaction centers and the light-harvesting complex of higher plants LHC-II with PCM reported an interesting exponential distance dependence [36,50], whereas Poisson-TrEsp calculations of excitonic couplings in photosystem I (PSI) did not reveal any obvious distance-dependence of the screening factor [40]. A recent study [49] ruled out that methodological differences between Poisson-TrEsp and PCM are responsible for the different results. The present work aims at re-investigating the excitonic couplings in PSI trimers (Figure 1), in order to resolve the above puzzle.

Whereas in the Poisson-TrEsp calculations [40], a many-cavity model was chosen, in which the pigments are treated as non-polarizable, the PCM calculations [36,50] used a two-cavity model. In the latter only the two pigments, for which the excitonic coupling is calculated, are treated by non-polarizable cavities, whereas the remaining pigments are considered to be part of the polarizable environment. An illustration of these models is shown in Figure 2.

There are arguments for both models. On one hand, the higher excited states of Chl are energetically not so far away from the first to treat them as part of the polarizable continuum and an explicit treatment in an extended exciton Hamiltonian might be more appropriate. On the other hand, standard exciton Hamiltonians, usually neglect the higher excited states of the pigments, and so an inclusion as part of the dynamic polarizability of the environment might be appropriate. Here we will investigate the difference between the two- and the many-cavity models, concerning the resulting screening factor.

Alternatively, the choice of pigment pairs could be responsible for the different results. As shown in the lower half of Figure 1 there is a broad distribution of mutual orientations of pigments in PSI. Here, we quantify this orientation by the orientational factor
(1)$$\kappa_{mn}={\vec{e}}_{m}\cdot {\vec{e}}_{n}-3\left({\vec{e}}_{m}\cdot {\vec{e}}_{mn}\right)\left({\vec{e}}_{n}\cdot {\vec{e}}_{mn}\right)\;$$
where ${\vec{e}}_{m}$ and ${\vec{e}}_{n}$ are unit vectors along the transition dipole moments ${\vec{\mu }}_{m}$ and ${\vec{\mu }}_{n}$ of pigments *m* and *n*, respectively, and the unit vector ${\vec{e}}_{mn}$ is oriented along the vector ${\vec{R}}_{m}-{\vec{R}}_{n}$ connecting the pigment centers. This orientational factor is well-known from the point-dipole approximation of the excitonic coupling. In the PCM study [36,50], a much smaller number of pigment pairs has been investigated than in the Poisson-TrEsp study [40]. Therefore, it could well be that in the latter, the exponential distance dependence of the screening factor is obtained for a subset of pigments and that the respective screening factors are masked by the scattered distribution function of screening factors obtained for the remaining pigment pairs. There is some evidence from an earlier analytical treatment using two point dipoles in a common spherical cavity by Hsu et al. [33] that the screening factor indeed depends on the mutual orientation of the two dipoles. If the two dipoles are oriented in-line, an increase of the excitonic coupling by the dielectric environment was reported, whereas for sandwich geometry the excitonic coupling was found to decrease. This effect was qualitatively explained by image dipoles, defined such as to represent the polarization of the dielectric. The polarization-mediated excitonic coupling then results from the interaction between the transition dipole of one pigment with the image dipole of the other. Whereas the image dipole of a dipole pointing at right angle onto a dielectric surface is oriented parallel to the original dipole, that of a dipole parallel to the dielectric surface is oriented antiparallel thereby enhancing and decreasing, respectively, the direct coupling between the two original dipoles (without the dielectric) [33]. But no distance dependence has been reported in this study.

The remaining of this work is organized in the following way. We start with a short review and extension of our previous derivation of the Poisson-TrEsp method [40]. The extension concerns the fact that now the dynamical polarizability of the environment is explicitly obtained [51], justifying the use of the optical dielectric constant in the Poisson-TrEsp calculations. We apply the two- and the many-cavity variant of Poisson-TrEsp to PSI trimers and determine the screening factors of the excitonic couplings. Next, we sort these screening factors according to the underlying geometry of transition dipole moments. Afterwards, we extract an empirical screening factor that depends on, both, inter-pigment orientation and distance. Finally the results are discussed, including model calculations for different geometries, more approximate cavities and using a point-dipole or extended-dipole approximation for the transition density of the pigments. We also relate our results for the screening factor to analytical expressions on simple model systems containing spherical cavities and find an interesting error compensation for one of those models, which allows this model to provide a quantitative description of the screening factor obtained for large interpigment distances with Poisson-TrEsp.

## 2. Theory

### Poisson-TrEsp Method and Screening Factor

The electronic part of the Hamiltonian of singly excited states of a pigment-protein complex can be expressed as
(2)$$H_{\mathrm{exc}}=\sum_{m}E_{m}\left|m><m\right|+\sum_{m,n}V_{mn}\left|m><n\right|\;,$$
where |m> denotes an excited state localized at pigment *m*, $E_{m}$ is the local excitation energy (site energy) of this state, and the excitonic coupling $V_{mn}$ describes the coupling between excited states of the complex localized at different pigments *m* and *n*.

For a perturbative treatment of the coupling of a pigment dimer to its environment, we consider a homodimer with equal site energies $E_{m}=E_{n}=E_{0}$ and a direct excitonic coupling $V_{mn}^{\left(0\right)}$. In the following, we want to investigate how this coupling changes if the excitonic coupling to a high-energy environment is taken into account.

For the homodimer, the delocalized eigenstates (exciton states) are obtained as
(3)$$|\pm >=\frac{1}{\sqrt{2}}(|m>\pm |n>)\;,$$
with energies $E_{\pm }^{\left(0\right)}=E_{0}\pm V_{mn}^{\left(0\right)}$. In the following, the influence of the excitonic coupling to the off-resonant high-energy transitions between the ground state and the *c*th excited state of the environmental building blocks *k*
(4)$$\hat{V}=\sum_{k}\sum_{c}\left(V_{10,0c}^{\left(m,k\right)}\left|m><k,c\right|+V_{10,0c}^{\left(n,k\right)}\left|n><k,c\right|\right)+h.c.$$
are studied. Here “*h.c.*” denotes the hermitian conjugate and |k,c> denotes a singly excited state, where environmental site *k* is in its *c*th excited state and all other building blocks are in their electronic ground state. The matrix element $V_{10,0c}^{\left(m,k\right)}$ contains the Coulomb coupling between the transition density $\rho_{10}^{\left(m\right)}\left(r_{m}\right)$ of the 0→1 transition of pigment *m* and the transition density $\rho_{0c}^{\left(k\right)}\left(r_{k}\right)$ of the 0→c transition of environmental building block *k*
(5)$$V_{10,0c}^{\left(m,k\right)}=\int dr_{m}\int dr_{k}\frac{\rho_{10}^{\left(m\right)}\left(r_{m}\right)\rho_{0c}^{\left(k\right)}\left(r_{k}\right)}{|r_{k}-r_{m}|}\;.$$

With a second-order perturbation theory in $\hat{V}$, the energies of the exciton states |±> are obtained as
(6)$$\begin{array}{c} E_{\pm }=E_{\pm }^{\left(0\right)}+\sum_{k}\sum_{c}\frac{{\left|<\pm |V|k,c>\right|}^{2}}{E_{\pm }^{\left(0\right)}-E_{c}^{\left(k\right)}}\\ =E_{0}\pm V_{mn}^{\left(0\right)}+\frac{1}{2}\sum_{k,c}\frac{V_{10,0c}^{\left(m,k\right)}V_{0c,10}^{\left(m,k\right)}+V_{10,0c}^{\left(n,k\right)}V_{0c,10}^{\left(n,k\right)}\pm V_{10,0c}^{\left(m,k\right)}V_{0c,10}^{\left(n,k\right)}\pm V_{10,0c}^{\left(n,k\right)}V_{0c,10}^{\left(m,k\right)}}{E_{0}\pm V_{mn}^{\left(0\right)}-E_{c}^{\left(k\right)}}\;. \end{array}$$

Taking into account that
(7)$$E_{0}\pm V_{mn}^{\left(0\right)}-E_{c}^{\left(k\right)}\approx E_{0}-E_{c}^{\left(k\right)}\;,$$
the difference between perturbed exciton energies becomes
(8)$$E_{+}-E_{-}=2V_{mn}^{\left(0\right)}+2\sum_{k,c}\frac{V_{10,0c}^{\left(m,k\right)}V_{0c,10}^{\left(n,k\right)}}{E_{0}-E_{c}^{\left(k\right)}}\;,$$
where we have also used the fact that the eigenfunctions of all states are assumed to be real-valued. Identifying the perturbed excitonic coupling $V_{mn}$ as half the splitting between eigenstates, results in
(9)$$V_{mn}=V_{mn}^{\left(0\right)}+\sum_{k}V_{mn}^{\left(k\right)}\;,$$
where $V_{mn}^{\left(0\right)}$ is the direct excitonic coupling between the pigments and
(10)$$V_{mn}^{\left(k\right)}=\sum_{c}\frac{V_{10,0c}^{\left(m,k\right)}V_{0c,10}^{\left(n,k\right)}}{E_{0}-E_{c}^{\left(k\right)}}$$
contains a superexchange-type contribution involving excitonic couplings to off-resonant states of the environmental building block *k*. The Coulomb coupling $V_{10,0c}^{\left(m,k\right)}=V_{0c,10}^{\left(m,k\right)}$ between transition densities (Equation (Equation 5)) is approximated by a sum over pairwise Coulomb interactions between atomic transition charges, known as TrEsp method [35],
(11)$$V_{10,0c}^{\left(m,k\right)}\approx \sum_{I,J}\frac{q_{I}^{\left(m\right)}\left(1,0\right)q_{J}^{\left(k\right)}\left(0,c\right)}{|R_{I}^{\left(m\right)}-R_{J}^{\left(k\right)}|}\;,$$
where the atomic transition charge $q_{I}^{\left(m\right)}\left(1,0\right)$ is placed at the *I*th atom of pigment *m*. These charges are obtained from a fit of the electrostatic potential of the ab-initio transition density. In order to relate the coupling $V_{mn}^{\left(k\right)}$ to the molecular polarizabilities of the environment, we apply a dipole approximation to the environmental building block *k* in Equation (Equation 11) resulting in
(12)$$V_{10,0c}^{\left(m,k\right)}=\sum_{I}\frac{q_{I}^{\left(m\right)}\left(1,0\right)d_{0c}^{\left(k\right)}\cdot \left(R_{I}^{\left(m\right)}-R_{k}\right)}{|R_{I}^{\left(m\right)}-R_{k}{|}^{3}}\;,$$
with the transition dipole moment of the 0→c transition of building block *k*
(13)$$d_{0c}^{\left(k\right)}=\sum_{J}q_{J}^{\left(k\right)}\left(0,c\right)R_{J}^{\left(k\right)}\;.$$

With the above approximations the environmental mediated excitonic coupling $V_{mn}^{\left(k\right)}$ in Equation (Equation 10) can be expressed as
(14)$$V_{mn}^{\left(k\right)}=\frac{1}{2}\sum_{I,J}q_{I}^{\left(m\right)}\left(1,0\right)q_{J}^{\left(n\right)}\left(0,1\right)\frac{\left(R_{J}^{\left(n\right)}-R_{k}\right){\hat{\alpha }}_{k}\left(-E_{0}\right)\left(R_{k}-R_{I}^{\left(m\right)}\right)}{|R_{J}^{\left(n\right)}-R_{k}{|}^{3}{\left|R_{I}^{\left(m\right)}-R_{k}\right|}^{3}}\;,$$
where we introduced the polarizability tensor of the *k*th building block at energy $E_{0}$ with Cartesian components
(15)$${\left(\alpha_{k}\left(-E_{0}\right)\right)}_{ij}=2\sum_{c}\frac{{\left(d_{0c}^{\left(k\right)}\right)}_{i}{\left(d_{0c}^{\left(k\right)}\right)}_{j}}{E_{c}^{\left(k\right)}-E_{0}}\;.$$

With this polarizability tensor Equation (Equation 14) can be interpreted in the following way. The transition charge $q_{I}^{\left(m\right)}\left(1,0\right)$ of pigment *m* creates a field
(16)$$E_{I}=q_{I}^{\left(m\right)}\left(1,0\right)\left(R_{k}-R_{I}^{\left(m\right)}\right)/{\left|R_{I}^{\left(m\right)}-R_{k}\right|}^{3}\;,$$
which induced a dipole moment in the *k*th environmental building block
(17)$$p_{k}={\hat{\alpha }}_{k}\left(-E_{0}\right)E_{I}\;,$$
that interacts with the partial charge $q_{J}^{\left(n\right)}\left(0,1\right)$ at position $R_{J}^{\left(n\right)}$ of pigment *n* via the dipole potential
(18)$$\phi_{p_{k}}\left(R_{J}^{\left(n\right)}\right)=p_{k}\left(R_{J}^{\left(n\right)}-R_{k}\right)/{\left|R_{J}^{\left(n\right)}-R_{k}\right|}^{3}\;.$$

Noting that the polarizability tensor ${\hat{\alpha }}_{k}\left(\hbar \omega \right)$ is related to the dynamic polarizability ${\hat{\alpha }}_{k}^{\mathrm{dyn}}\left(\hbar \omega \right)$, that describes the polarization by a field of frequency ω by [51,52]
(19)$$\begin{array}{c} {\hat{\alpha }}_{k}^{\mathrm{dyn}}\left(\hbar \omega \right)=\frac{1}{2}\left({\hat{\alpha }}_{k}\left(\hbar \omega \right)+{\hat{\alpha }}_{k}\left(-\hbar \omega \right)\right)\approx \frac{1}{2}{\hat{\alpha }}_{k}\left(-\hbar \omega \right) \end{array}$$
we can identify the polarization in Equation (Equation 14) as a fast (optical) polarization of the environment, which takes care of the fact that an electronic excitation energy transfer event does not leave any time for a slow polarization of the environment.

The above derivation is exploited in the Poisson-TrEsp method [40,53]. Please note, that in our earlier derivation a too drastic approximation for the energy difference in Equation (Equation 7) was used that led to the static rather than the dynamic polarizability of the environment, which was interpreted, however, as dynamical polarizability by using the physical argument that the energy transfer is fast compared to nuclear motion [40]. Here, a more rigorous foundation of this argument is provided.

In the Poisson-TrEsp method, the protein/solvent environment is modeled as a homogeneous dielectric with optical dielectric constant ϵ. The transition charges of the pigments are placed in molecule-shape cavities with ϵ=1 inside the cavities and optical dielectric constant $\epsilon =n^{2}$ outside, where *n* is the (average) refractive index. Please note that in our treatment we distinguish between the two-cavity and the many-cavity model, as described in the introduction. Whereas in the two-cavity model there are only cavities for those two pigments for which the coupling is determined, in the many-cavity model also the cavities of the remaining pigments are included (Figure 2). The electrostatic potential of the transition density of chromophore *m*, $\phi_{m}^{\left(1,0\right)}\left(r\right)$ is obtained by solving a Poisson equation
(20)$$\nabla \left(\epsilon \left(r\right)\nabla \phi_{m}^{\left(1,0\right)}\left(r\right)\right)=-4\pi \sum_{I}q_{I}^{\left(m\right)}\left(1,0\right)\delta \left(r-R_{I}^{\left(m\right)}\right)\;,$$
where ϵ(r) equals one if r points into a cavity and $n^{2}$ otherwise. The coupling between chromophores *m* and *n* is then obtained as
(21)$$V_{mn}=\sum_{J}\phi_{m}^{\left(1,0\right)}\left(R_{J}^{\left(n\right)}\right)q_{J}^{\left(n\right)}\left(1,0\right)\;.$$

By comparing the coupling $V_{mn}$, obtained with Poisson-TrEsp, with the direct interaction $V_{mn}^{\left(0\right)}$, the screening factor
(22)$$f_{mn}=\frac{V_{mn}}{V_{mn}^{\left(0\right)}}$$
results. The principal aim of the present work is to study the dependence of $f_{mn}$ on the interpigment distance $R_{mn}$ and orientation $\kappa_{mn}$ (Equation (Equation 1)).

The Poisson equation is solved with a finite difference method using the program MEAD [54,55]. The atomic transition charges were obtained from a fit of the electrostatic potential of the ab-initio transition density of geometry-optimized Chl *a*. Details of the quantum chemical calculations and the numerical values of the transition charges are given in ref. [35]. (We used the charges obtained with the B3LYP exchange correlation functional, however, this choice is not critical.) The average refractive index *n* of PSI has been estimated [56] based on a comparison of the integral oscillator strength of protein-bound and solvent extracted pigments, measured in ref. [57], as n=1.39±0.04, which leads to an optical dielectric constant $n^{2}$ in the range 1.82–2.04. In the present calculations we use $n^{2}=2$.

## 3. Results

### 3.1. Screening Factors of PSI Trimer in Many-Cavity versus Two-Cavity Model

The screening function of inter-pigment excitonic couplings of all pigment pairs of PSI trimers with intermolecular distances smaller than 42 Å has been evaluated. A comparison of the screening factors $f_{mn}$ as a function of intermolecular distances $R_{mn}$, obtained in the many-cavity and the two-cavity model is presented in Figure 3. The screening factors in the two-cavity model are somewhat less scattered, due to the more homogeneous environment, but in both models there is no obvious distance dependence of the screening factors visible. Close inspection of the two-cavity results seem to suggest that a subset of points at short intermolecular distances could lie on an exponential curve. We will try to discriminate these points by sorting the pigment pairs according to their mutual pigment orientations in the following.

### 3.2. Dependence of Screening Factors on Mutual Orientation and Distance

The mutual orientation of pigments in the different pairs is characterized by the orientational factor $\kappa_{mn}$ of the point-dipole coupling (Equation (Equation 1)). Examples of dipole orientations and respective $\kappa_{mn}$ values are given in Table 1. Particular orientations of interest will be those for which the point-dipole excitonic coupling vanishes ($\kappa_{mn}=0$), the “sandwich” geometry ($|\kappa_{mn}|=1$) and the “in-line” geometry ($|\kappa_{mn}|=2$).

We divide the pigment pairs of PSI into three groups according to the ranges of the $|\kappa_{mn}|$ values given in Table 2.

Group 1 is expected to contain the pigment pairs with small excitonic couplings, whereas the pairs with sandwich and in-line type geometry are in groups 2 and 3, respectively. The screening factors of the pigments in the three groups are shown in Figure 4 as a function of interpigment distance and color-coded with respect to the vacuum coupling values. The pigment pairs in the first group exhibit no systematic distance dependence and a large scattering of the screening factor values. In groups 2 and 3 the scattering of data points is much less than in group 1 and the distance dependencies of the screening factors are best described by a distance-independent function $f_{mn}^{\left(2\right)}$ (Equation (Equation 23)) and an exponential screening function $f_{mn}^{\left(3\right)}\left(R_{mn}\right)$ (Equation (24)), respectively, with
(23)$$\begin{array}{ccc} f_{mn}^{\left(2\right)} & = & 0.60\;\mathrm{and} \end{array}$$
(24)$$\begin{array}{ccc} f_{mn}^{\left(3\right)}\left(R_{mn}\right) & = & 0.59+39.6\exp\{-0.56R_{mn}/Å\}\;. \end{array}$$

The larger scattering of data points obtained with the many-cavity model in Figure 3 translates into a larger scattering of the data points of the individual groups in Figure 4, but the same fit functions can be used for, both, the two-cavity and the many-cavity model. In order to quantify the quality of the approximate screening factor, we calculated the variance between the Poisson-TrEsp couplings $V_{mn}^{\left(P-\mathrm{TrEsp}\right)}$ and the approximate coupling obtained by multiplying the vacuum coupling $V_{mn}^{\left(\epsilon =1\right)}$ by an approximate screening factor $f_{mn}^{\left(\mathrm{appr}\right)}$. Hence the variance reads
(25)$$\sigma^{2}=\frac{1}{N_{\mathrm{pairs}}}\sum_{m,n}{\left(V_{mn}^{\left(P-\mathrm{TrEsp}\right)}-f_{mn}^{\left(\mathrm{appr}\right)}V_{mn}^{\left(\epsilon =1\right)}\right)}^{2}$$
where $N_{\mathrm{pairs}}$ is the number of pigment pairs included in the sum. In addition to the $f_{mn}^{\left(\mathrm{appr}\right)}$ obtained from the fit of of the screening factors of the pigment pairs in groups 2 and 3, we have tested a unified function for the screening factor
(26)$$f_{mn}^{\left(4\right)}\left(R_{mn},\kappa_{mn}\right)=0.60+39.6\;\;\Theta (|\kappa_{mn}\left|-\kappa_{0}\right)\exp\left\{-0.56R_{mn}/Å\right\}$$
that contains the Heaviside step function
(27)$$\Theta (|\kappa_{mn}\left|-\kappa_{0}\right)=\left\{\begin{array}{c} 1\;|\kappa_{mn}|\geq \kappa_{0}\\ 0\;\mathrm{else} \end{array}\right.$$

For large distances or $|\kappa_{mn}|<\kappa_{0}$ a constant screening factor of 0.6 is taken, as determined from the fit of group 2 pigment pairs (Equation (Equation 23)), whereas for $|\kappa_{mn}|>\kappa_{0}$ we assume the exponential distance dependence of the group 3 pigment pairs (Equation (24)). The smallest variance for the pigment pairs of PSI is obtained by choosing
(28)$$\kappa_{0}=1.17$$
as shown in Figure 5. We note that the use of a Heaviside function in Equation (Equation 26) is a simplification, since the screening factors calculated with Poisson-TrEsp undergo a more smooth transition with respect to the mutual orientation of pigments, as will be discussed in more detail below with model calculations. The variances between the different approximate screening factors for the different groups and the Poisson-TrEsp couplings obtained in the two-cavity model are summarized in Table 3.

As expected, the constant screening factor $f_{mn}^{\left(\mathrm{appr}\right)}=f_{mn}^{\left(2\right)}=0.6$ gives the lowest variance for the group 1 and group 2 pigment pairs, whereas the exponential screening factor $f_{mn}^{\left(\mathrm{appr}\right)}=f_{mn}^{\left(3\right)}\left(R_{mn}\right)$ works best for group 3. The geometry-dependent screening factor $f_{mn}^{\left(4\right)}\left(R_{mn},\kappa_{mn}\right)$ combines the advantages of the other two factors and provides a good description for all groups. The variance obtained with $f_{mn}^{\left(4\right)}\left(R_{mn},\kappa_{mn}\right)$ for all pigment pairs is almost an order of magnitude smaller than that obtained with the exponential screening factor and still a factor 3–4 smaller than that obtained with a constant screening factor.

The variances obtained in the many-cavity model are somewhat larger but show the same qualitative behavior (Appendix A).

## 4. Discussion

Interestingly, an exponential distance dependence of the screening factor is only found for in-line-type dipole geometries (group 3 in Table 2 and Figure 4), whereas sandwich-type-geometries (group 2 in Table 2 and Figure 4) show a much weaker distance dependence of the screening factor that can be approximated by a constant value of 0.6, which is also obtained at large distances for the group 3 pigment pairs. Hence, the puzzle is solved, the different results in the earlier PCM [36,50] and Poisson-TrEsp [40] studies were obtained due to a different choice of pigment orientations. Practically all pigment pairs contributing to the exponential distance dependence of the screening factor in the earlier PCM work [36,50] contain a group 3 (in-line type) geometry. Moreover, in these studies, additional pigment pairs were created by translating one pigment along the center-to-center vector (${\vec{e}}_{mn})$ in Equation (Equation 1), a translation, which conserves the orientation factor $\kappa_{mn}$. Two prominent examples of this study, the P_D1_-P_D2_ pair of the reaction center of photosystem II and the Chl_602_-Chl_607_ dimer of the LHC-II light-harvesting antenna of plants were investigated with Poisson-TrEsp and show the expected exponential distance dependence of the screening factor (Appendix A).

There are two well-known analytical models that both approximate the pigment and its environment as a spherical cavity with a point-transition dipole $d_{0}$ in the center and a homogeneous dielectric with optical dielectric constant $n^{2}$ outside. From electrostatic theory, it is well known that the electrostatic potential φ(r) in the medium (that is, outside the cavity) is given as [58]
(29)$$\varphi \left(r\right)=\frac{3}{2n^{2}+1}\frac{d_{0}\cdot r}{r^{3}}\;,$$
where r is the vector from the center of the cavity to a point outside the cavity. A critical point here is that in the solution of the Poisson equation for φ(r) the presence of the second cavity (around the second dipole) is neglected.

In one model, this cavity is implicitly taken into account by mapping the two-cavity system onto a system of two interacting effective transition dipoles without cavities in a dielectric medium, as illustrated in the upper and middle panels of Figure 6. The effective dipole moment $d_{\mathrm{eff}}=\frac{3n^{2}}{2n^{2}+1}d_{0}$ is chosen such that the potential $\varphi_{\mathrm{eff}}\left(r\right)=\frac{1}{n^{2}}\frac{d_{\mathrm{eff}}\cdot r}{r^{3}}$ equals that in Equation (Equation 29). The interaction between the two effective transition dipoles in the dielectric medium (middle panel of Figure 6) experiences a screening by the usual $1/n^{2}$ factor, resulting in an overall screening factor [33,59,60]
(30)$$f_{L}=\frac{1}{n^{2}}{\left(\frac{3n^{2}}{2n^{2}+1}\right)}^{2}=\frac{9n^{2}}{{\left(2n^{2}+1\right)}^{2}}$$
of the excitonic coupling, first derived by Agranovich and Galanin [59] and related to the Lorentz local field factor.

In the alternative model, the second cavity is completely neglected (lower panel of Figure 6), and the potential of the first dipole (with cavity) in the dielectric medium is expressed as a sum of a vacuum and a medium contribution $\varphi \left(r\right)=\varphi^{\left(0\right)}\left(r\right)+\varphi^{\left(1\right)}\left(r\right)$, with $\varphi^{\left(0\right)}\left(r\right)=\frac{d_{0}\cdot r}{r^{3}}$ and $\varphi^{\left(1\right)}\left(r\right)=\frac{2(\epsilon -1)}{2\epsilon +1}\frac{d_{0}\cdot r}{r^{3}}$. These two contributions are proportional to the direct (vacuum) and the indirect (mediated by the medium) couplings, $V^{\left(0\right)}$ and $V^{\left(1\right)}$, respectively, of the dipole in the cavity and the dipole without cavity. Hence, the screening factor $f_{c}=\left(V^{\left(0\right)}+V^{\left(1\right)}\right)/V^{\left(0\right)}$ is obtained as [33,49,50]
(31)$$f_{c}=\frac{3}{2n^{2}+1}\;.$$

For the present $n^{2}=2$, we obtain $f_{L}=0.72$ and $f_{c}=0.6$. Interestingly, $f_{c}=0.6$ describes the screening of the excitonic couplings in the group 2 and group 3 pigment pairs of PSI at large distances, whereas the alternative factor $f_{L}=0.72$ deviates. This result is somewhat surprising, since $f_{L}$ takes into account the second cavity by the mapping procedure described above, whereas $f_{c}$ does not. As will be shown below, this result relies on an error compensation effect in $f_{c}$.

In contrast to the well-behaved screening factors of group 2 and group 3 pigment pairs, the screening factors of the group 1 pigment pairs in Figure 4 (upper panel) hardly show any systematic distance dependence. Large scattering is observed around the constant value of 0.6 for all distances. Due to the small orientational factor of the group 1 pairs, the excitonic couplings are rather small and, hence, one may have the suspicion that the scattering is related to singularities occurring for small vacuum couplings. This point is studied in more detail in the model calculations, described in the following.

### Model Calculations

We consider the model chlorophyll *a* dimer systems shown in Figure 7. In order to study the distance dependence of the screening factor, we displace one pigment with respect to the other. If the displacement is performed along the center-to-center vector connecting the two pigments (as denoted by the purple arrows in Figure 7), the geometry factor $\kappa_{mn}$ does not change and we obtain the screening factors shown in the upper part of Figure 8. As expected, the screening factor of the in-line dimer exhibits an exponential distance dependence, whereas that of the sandwich dimer is much less distance-dependent.

In order to investigate the change of the screening factor with varying orientation for fixed interpigment distance (in this case 12 Å), we have rotated one of the two pigments in the in-line and sandwich dimers around an axis perpendicular to their transition dipole moment (Figure 7). As seen in the lower left panel in Figure 8, for the sandwich dimer the screening factor practically stays constant close to 0.6, in agreement with the empirical screening function $f_{mn}^{\left(4\right)}$ (Equation (Equation 26)) since $|\kappa_{mn}|<\kappa_{0}$ for all rotation angles φ of the sandwich dimer (Appendix A). For the in-line dimer, we have $|\kappa_{mn}|<\kappa_{0}$ for $|\varphi |>54^{{}^{\circ}}$ (Appendix A), and the screening factor is roughly constant close to 0.6 (The changes for very large φ are not critical, since the couplings are small for these angles.) In contrast, for $|\varphi |<54^{{}^{\circ}}$ the screening factor steadily increases until it reaches a maximum value of 0.8 at φ=0. Our empirical screening function $f_{mn}^{\left(4\right)}$ (Equation (Equation 26)) at $\left|\varphi \right|=54^{{}^{\circ}}$ makes a jump but stays constant for smaller angle magnitudes $|\varphi |<54^{{}^{\circ}}$, as shown by the black-dashed line in the lower left panel of Figure 8. Hence there is a quantitative difference between the Poisson-TrEsp results and the empirical screening function, but the qualitative behavior goes in the same direction.

If we displace the upper pigment in the sandwich dimer in the lower panel of Figure 7 in a horizontal direction with respect to the center-to-center line (as indicated by the yellow arrow in this figure), the orientational factor is changing, in addition to the interpigment distance. The respective screening constants are compared in the lower right part of Figure 8 with the couplings obtained in vacuum and in the dielectric medium. Interestingly, singularities of the screening factor are observed in the neighborhood of the zero crossing of the vacuum excitonic coupling $V_{mn}^{\left(0\right)}$, due to the slightly shifted zero crossing of the medium excitonic coupling $V_{mn}$. Note that the screening factor is defined as the ratio between the two (Equation (Equation 22)). Outside a ± 5 cm^−1^ coupling window around the zero medium coupling, the screening factor takes already values between 0.55 and 0.65, which explains why most of the scattered screening factors of the group 1 pigment pairs in PSI have a small coupling (Figure 4, upper panels). Since in pigment-protein complexes, the energy transfer and the optical properties are determined by the strong couplings, that is the pigment pairs in group 2 and 3, we can afford to approximate the screening factors of the group 1 pigment pairs by a constant. However, in general, in the absence of strong couplings, care has to be taken in the treatment of the dielectric screening of pigment pairs with a small orientational factor $\kappa_{mn}$. Please note also that at short distances, there are some pigment pairs in group 1 and group 2 (Figure 4, upper and middle panels) with intermediate coupling strengths, where the screening factor varies between 0.4 and 0.8. These variations cannot be predicted by our empirical screening function, since they rely on specific cavity overlap effects, that require a full Poisson-TrEsp calculation.

Finally, we would like to come back to the analytical screening factors, discussed above (Equations (Equation 30) and (Equation 31)) and demonstrate an error-compensation effect that explains why $f_{c}\left(n^{2}=2\right)=0.6$ is able to describe the screening value obtained for group 2 dimers and for group 3 dimers for large interpigment distances in PSI (Figure 4), whereas $f_{L}\left(n^{2}=2\right)=0.72$ gives a larger value. For this purpose we have numerically solved the Poisson equation for two point dipoles in two spherical cavities with a cavity radius of 5.8 Å and determined the screening constant as a function of intermolecular distance, shown as solid lines in Figure 9. For large distances, where the cavities do not overlap, a constant screening factor of 0.72 is obtained, which is identical to the analytical estimate by Agranovich and Galanin [59] (Equation (Equation 30) with $n^{2}=2$), despite the fact, that their mapping relied on the solution of the electrostatic potential of each dipole in its own cavity, without the cavity of the second dipole. As expected from the analytical model, if we neglect one spherical cavity completely, we obtain a screening factor of 0.6 (the dotted lines in Figure 4), which is close to the screening factors, obtained using a molecular cavity and atomic transition charges (Figure 8 upper part) or an extended dipole (extend 7.8 Å) (the dashed lines in Figure 9). Hence, in the model with just one spherical cavity, neglecting the second spherical cavity compensates the error that is caused by assuming a spherical rather than a molecular cavity. Neglecting one molecular cavity gives somewhat smaller screening factors around 0.55 for large distances (the dot-dashed curves in Figure 9) and a steeper rise of the screening factor for short distance and in-line geometry. The latter result highlights the importance of having both transition dipoles in a common cavity in order to obtain an increase of the screening factor. The transition towards this scenario, obviously occurs more sharply in the case of one molecular cavity (rather than two cavities). Please note also that the results obtained for extended dipoles and two molecular cavities (red and black dashed lines in Figure 8) are qualitatively close to those obtained using the actual atomic transition charges (upper panel of Figure 8) showing that the exact transition charge distribution is not so critical for the screening.

## 5. Conclusions

In the present work we have explained the differences between earlier PCM [36,50] and Poisson-TrEsp [40] results by differentiating between the mutual orientations of pigments. Based on this distinction, an empirical screening function is proposed that contains an exponential distance dependence for in-line type geometries and a constant screening for sandwich like dimers. For large interpigment distances the screening factor becomes independent on distance and for in-line and sandwich type geometries adopts a constant value of 0.6. This value can be qualitatively understood by analytical models considering point dipoles in spherical cavities. We have to note that the quantitative agreement of one such models relies on a fortuitous error compensation. At short intermolecular distances, where the molecular cavities start to overlap, different behavior is obtained for in-line and sandwich dimers. Qualitatively, a transition occurs towards a scenario discussed by Hsu et al. [33] for two point dipoles in a single spherical cavity in terms of image dipoles causing an enhanced and decreased coupling between in-line and between sandwich dipoles, respectively (see introduction). A third group of pigment geometries is identified, for which the screening factor exhibits singularities due to slightly different zero crossings of the vacuum and the medium excitonic couplings. Since there are many strong couplings, these small couplings most likely do not play any role for energy transfer and optical spectra of PSI, but could be more important for systems with unfavorable dipole geometries [62]. Finally, we note that the results obtained in the two- and in the many-cavity models are qualitatively very similar. In particular, the same empirical screening function is obtained in the two models. In general, the orientation and distance dependence of the screening factors in the many-cavity models is less smooth, because of the less homogeneous dielectric environment.

## Figures and Tables

**Figure 1 ijms-25-09006-f001:**
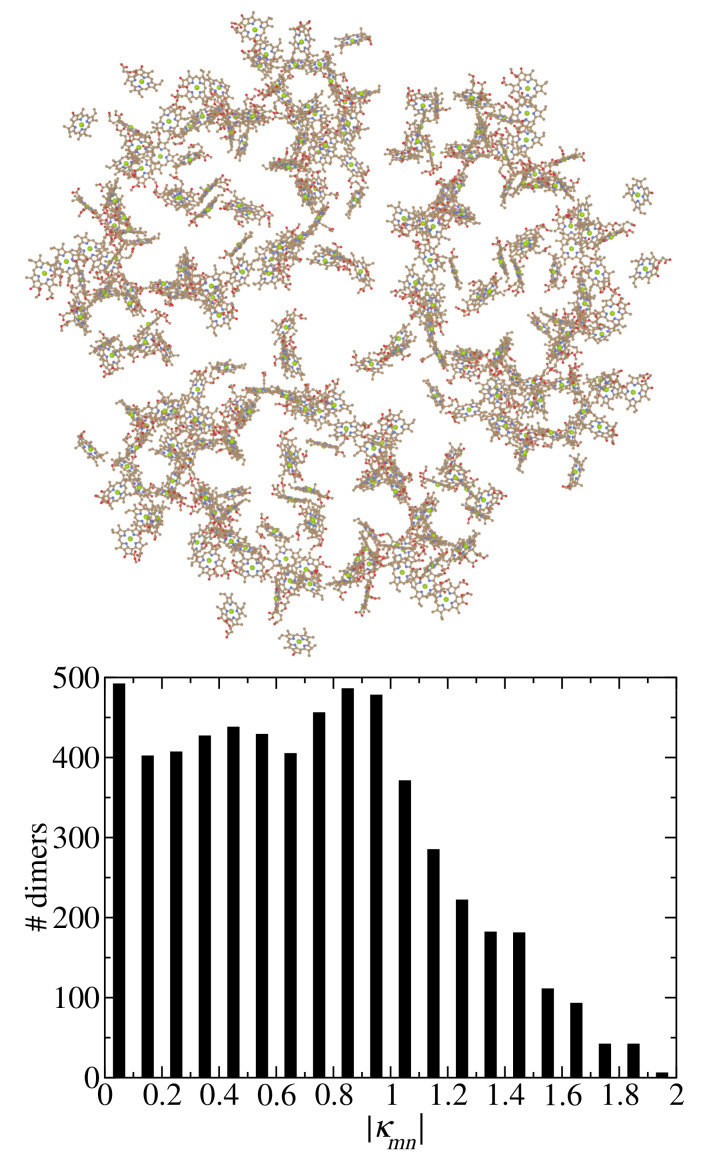
(**Upper part**): Chlorophyll pigments of PSI trimer [12]. (**Lower part**): Number of pigment pairs in PSI as a function of the absolute magnitude of their orientational factor $\kappa_{mn}$ (Equation (Equation 1)). Structure drawn with Blender [13].

**Figure 2 ijms-25-09006-f002:**
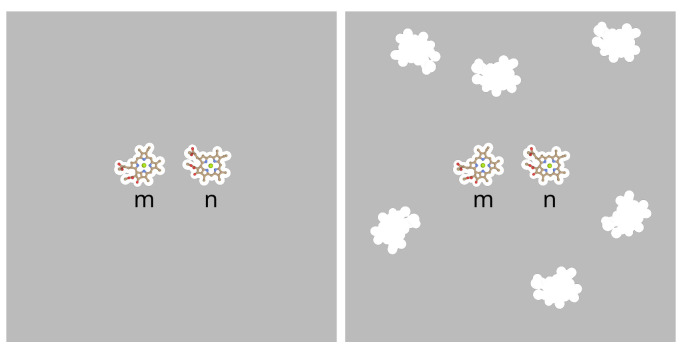
Illustration of two-cavity model (**left part**) and many-cavity model (**right part**). When the excitonic coupling between pigments *m* and *n* is calculated in the two-cavity model, the whole environment is being treated as a homogeneous dielectric. In the many-cavity model the remaining pigment cavities of the photosystem are treated as non-polarizable. For further explanation see text.

**Figure 3 ijms-25-09006-f003:**
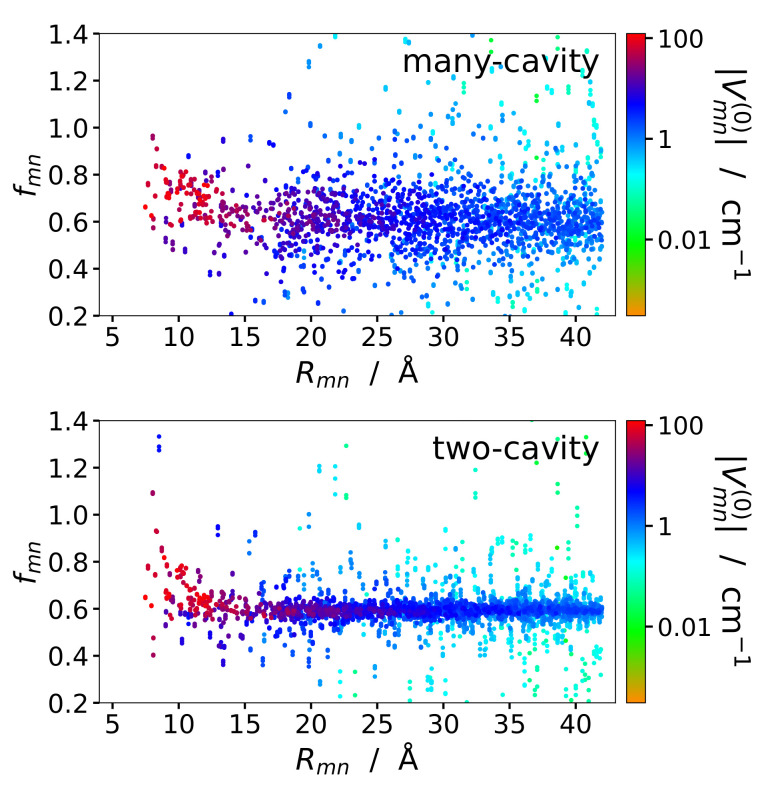
Screening factors $f_{mn}$ of excitonic couplings $V_{mn}$ between chlorophylls in PSI trimer as a function of center-to-center distance $R_{mn}$, obtained in the many-cavity model (**upper part**) are compared to those obtained in the two-cavity model (**lower part**). The points are color-coded according to their excitonic coupling $V_{mn}^{\left(0\right)}$ obtained without environment, as quantified in the legend.

**Figure 4 ijms-25-09006-f004:**
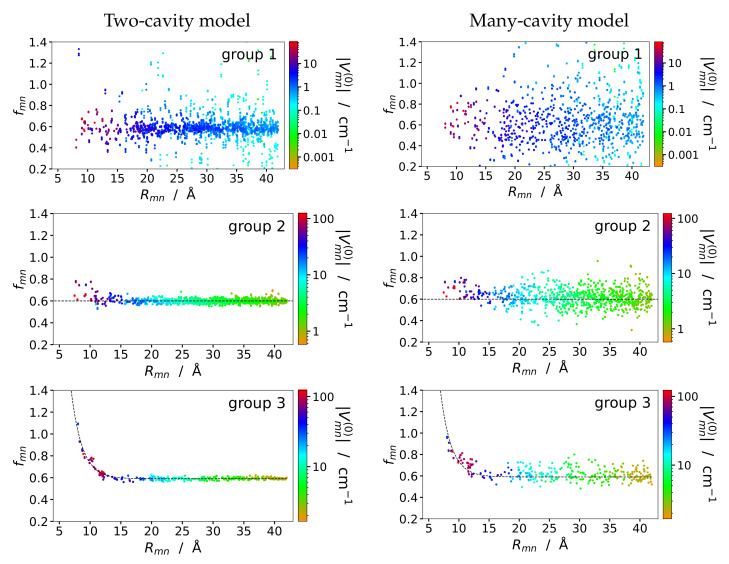
Same as in Figure 3, but sorted according to pigment orientations in the pairs, upper panels: group 1, middle panels: group 2, lower panels: group 3. Groups are defined in Table 2. The left and the right colums contain the results obtained in the two- and the many-cavity model, respectively (lower and upper panel in Figure 3). The black-dashed lines in the middle and lower panels describe a fit of the data with Equations (Equation 23) and (24), respectively.

**Figure 5 ijms-25-09006-f005:**
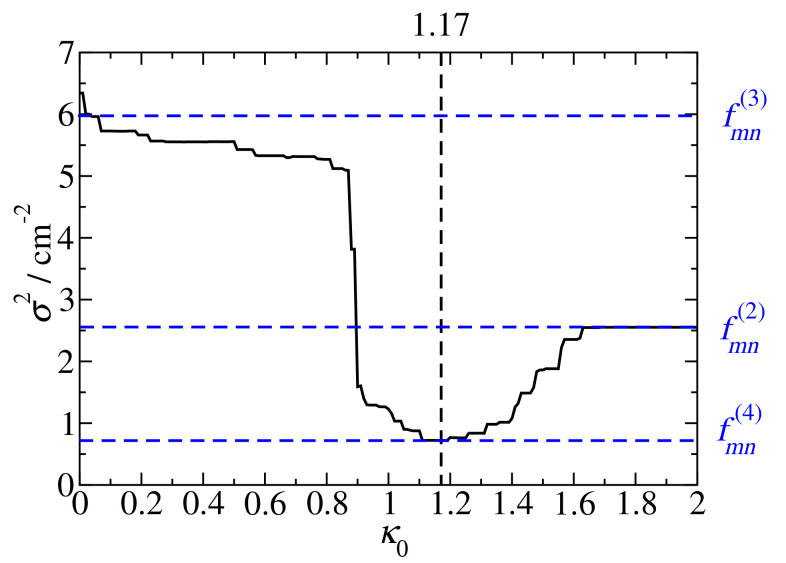
Variance $\sigma^{2}$ between Poisson-TrEsp coupling $V_{mn}^{\left(P-\mathrm{TrEsp}\right)}$ and the approximate coupling $V_{mn}^{\left(\epsilon =1\right)}f_{mn}^{\left(4\right)}$ (Equation (Equation 26)), obtained for different values $\kappa_{0}$. The lowest $\sigma^{2}$ value is obtained for $\kappa_{0}=1.17$, marked by a horizontal dashed line. This value is used in the screening function $f_{mn}^{\left(4\right)}$ in Equation (Equation 26), as indicated by the lowest horizontal blue-dashed line. The remaining screening functions $f_{mn}^{\left(2\right)}$ (Equation (Equation 23)) and $f_{mn}^{\left(3\right)}$ (Equation (24)) follow for the limiting situations $\kappa_{0}=2$ and $\kappa_{0}=0$, respectively, as shown by the middle and upper horizontal blue-dashed lines.

**Figure 6 ijms-25-09006-f006:**
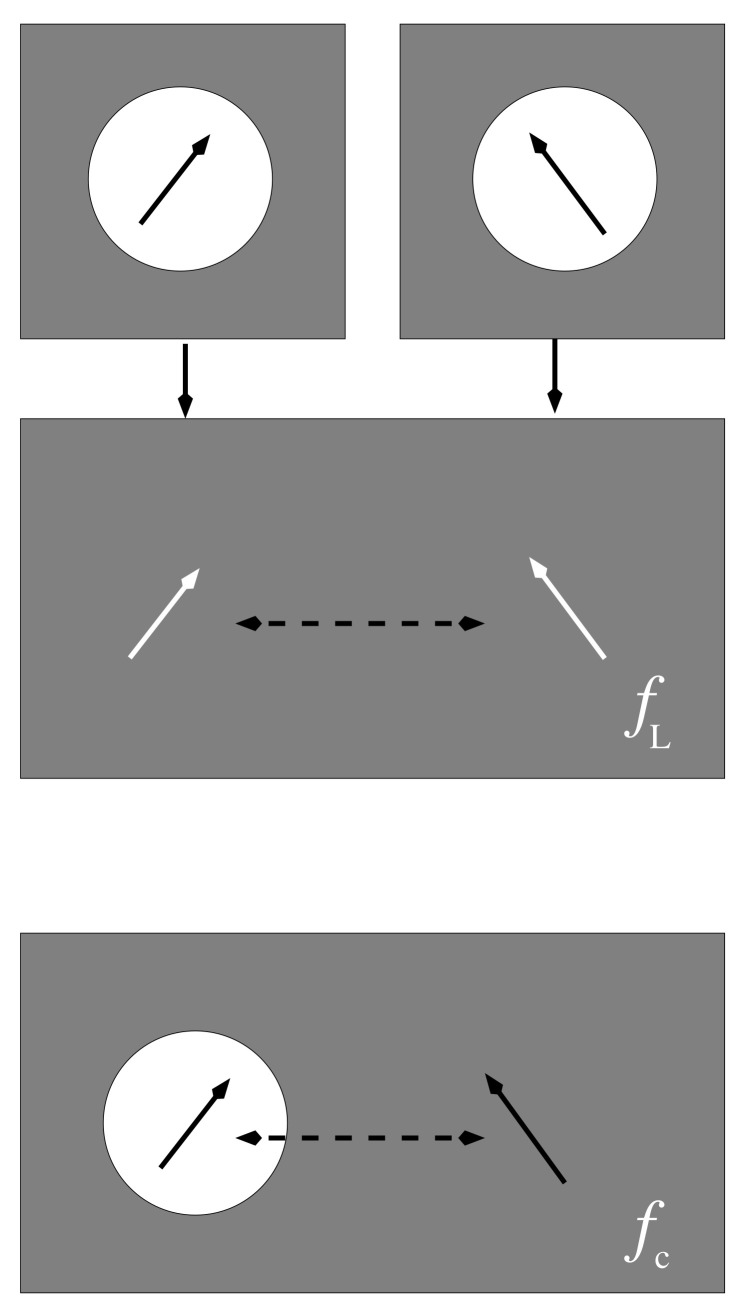
Illustration of two spherical-cavity models, with point-transition dipoles in the center, representing two pigments. In the first model (**upper and middle panels**) both dipoles (black arrows) are situated in empty spherical cavities, but in the solution of the Poisson equation for the electrostatic potential only one cavity is taken into account (**top panel**). The two cavity dipoles are then mapped onto two effective dipoles (white arrows) without cavity and the coupling of the latter in the dielectric medium is considered (**middle panel**), revealing the screening factor $f_{L}$ (Equation (Equation 30)). The second model neglects the spherical cavity of one pigments in, both, the solution of the Poisson equation and the calculation of the screening (**lower panel**), resulting in the screening factor $f_{C}$ (Equation (Equation 31)).

**Figure 7 ijms-25-09006-f007:**
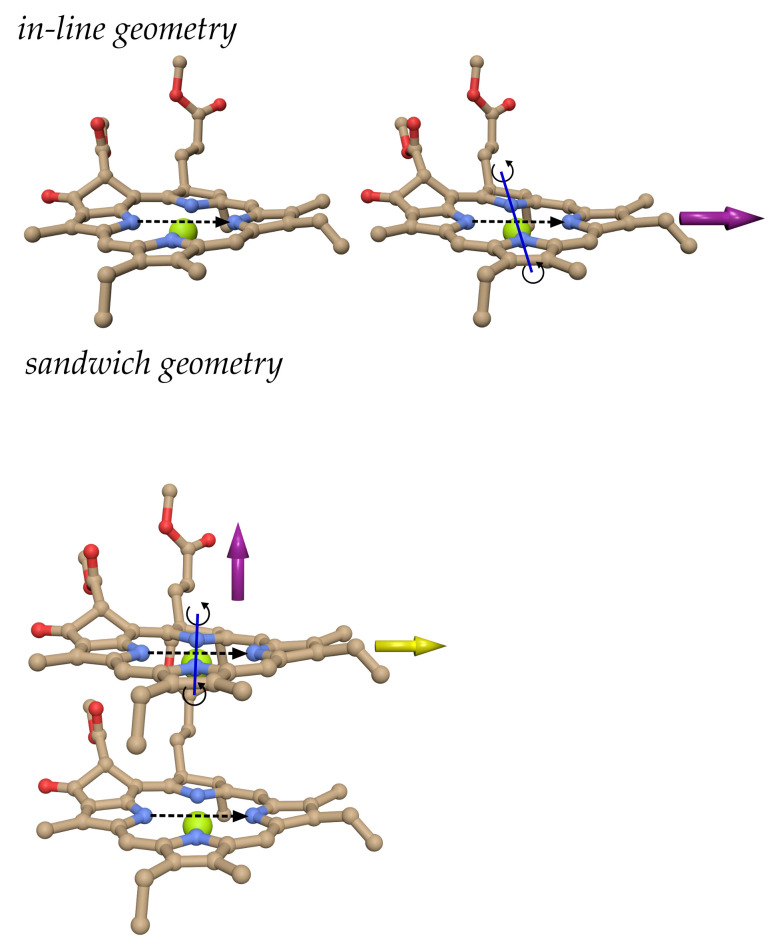
Two chlorophyll *a* pigments in in-line (**upper part**) and sandwich (**lower part**) configurations. The purple arrows indicate directions in which one pigment is translated in the model calculations without changing the orientational factor. The yellow arrow in the lower figure denotes an alternative translation direction, also investigated in the coupling calculations. The solid blue line in the right pigment of the upper part and the upper pigment in the lower part indicates a rotation axis, used to investigate further intermolecular orientations. The dashed black arrow in the center of the pigments indicates the direction of transition dipole moment.

**Figure 8 ijms-25-09006-f008:**
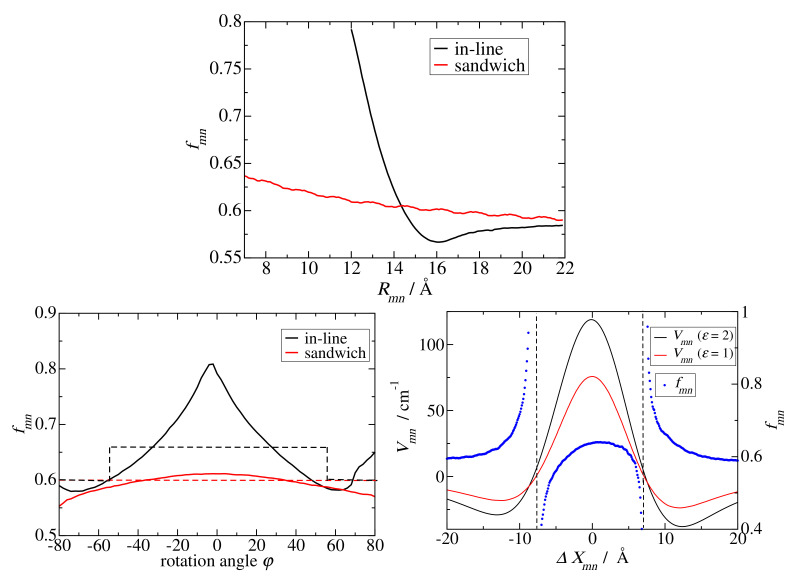
(**Upper part**): Screening factors $f_{mn}$ calculated for different configurations (center-to-center distances $R_{mn}$) of the model dimer, obtained from the sandwich and the in-line dimer in Figure 7 by displacing one pigment along the vector connecting the pigment centers (the violet arrows in Figure 7). (**Lower left part**): Same as in upper part but for conformations obtained by rotating one pigment by an angle φ around an axis perpendicular to its transition dipole moment (the blue line in the upper and lower part of Figure 7), where φ=0 refers to the original orientation. The dashed lines show the screening factor obtained by using the empirical expression in Equation (Equation 26). (**Lower right part**): Excitonic couplings in vacuum ($V_{mn}^{\left(0\right)}=V_{mn}\left(\epsilon =1\right)$, black line) and dielectric medium ($V_{mn}=V_{mn}\left(\epsilon =2\right)$, red line) and screening factors ($f_{mn}$, blue dots) calculated for different configurations, obtained from the lower dimer in Figure 7 by displacing the upper pigment along the yellow arrow by $\Delta X_{mn}$, where $\Delta X_{mn}=0$ corresponds to the sandwich geometry shown in this figure. The vertical dashed lines mark the positions of the singularities of $f_{mn}$, where $V_{mn}^{\left(0\right)}=0$ and $V_{mn}\neq 0$.

**Figure 9 ijms-25-09006-f009:**
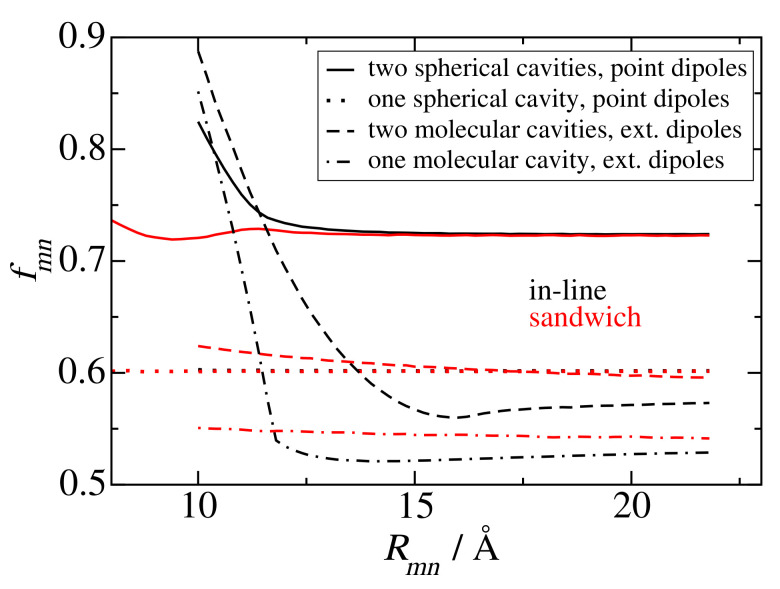
Screening factors $f_{mn}$ as a function of inter molecular distance, obtained for in-line (black curves) and sandwich (red curves) geometries of transition dipole moments, using different approximations for the shape and number of molecular cavities and the transition density. Solid lines are obtained for two spherical cavities with two point dipoles, dotted lines for one molecular cavity with one point dipole and another point dipole without cavity, dashed lines for two molecular cavities with extended dipoles, and dot-dashed lines for one extended dipole in a molecular cavity and another extended dipole without cavity. A dipole extend of 8.7 Å and a cavity radius of 5.8 Å were used, as inferred from electrostatic calculations of dispersive transition energy shifts on a related molecule (bacteriochlorophyll *a*) [61].

**Table 1 ijms-25-09006-t001:** Examples of different mutual orientations of transition dipole moments and resulting absolute magnitude of the orientation factors $\kappa_{mn}$ (Equation (Equation 1)).

Orientation	$|\kappa_{\mathrm{mn}}|$
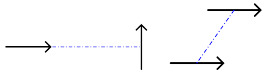	0
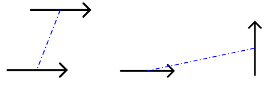	0.6
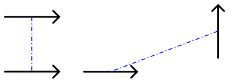	1
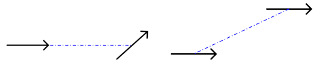	1.45
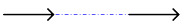	2

**Table 2 ijms-25-09006-t002:** Definition of groups used for the sorting of pigment pairs according to the mutual orientation of the pigments as quantified by absolute magnitude of the orientational factor $\kappa_{mn}$.

Group	Geometry
1	$0.0\leq \;|\kappa_{mn}|\;\leq 0.6$
2	$0.6\leq \;|\kappa_{mn}|\;\leq 1.2$
3	$1.2\leq \;|\kappa_{mn}|\;\leq 2.0$

**Table 3 ijms-25-09006-t003:** Variances $\sigma^{2}$ between Poisson-TrEsp couplings, obtained in the two-cavity model, and approximate couplings (Equation (Equation 25)) obtained for different $f_{mn}^{\left(\mathrm{appr}\right)}$. Low variances are high-lighted in bold style.

	$\sigma^{2}$ / ${\mathrm{cm}}^{-2}$		
**Group**	$f_{\mathrm{mn}}^{\left(2\right)}$ ^ **(a)** ^	$f_{\mathrm{mn}}^{\left(3\right)}\left(R_{\mathrm{mn}}\right)$ ^ **(b)** ^	$f_{\mathrm{mn}}^{\left(4\right)}\left(R_{\mathrm{mn}},\kappa_{\mathrm{mn}}\right)$ ^ **(c)** ^
1	**0.47**	2.64	**0.47**
2	**1.06**	11.38	**0.94**
3	12.90	**0.72**	**0.82**
all	2.55	6.00	**0.72**

^(a)^ Equation (Equation 23) (constant screening), ^(b)^ Equation (24) (exponential screening), ^(c)^ (Equation (Equation 26) (geometry-dependent screening).

## Data Availability

The numerical values of the excitonic couplings in vacuum ($V_{mn}^{\left(0\right)}$) and in the dielectric medium ($V_{mn}$) of PSI in the different groups in Figure 4 are provided in the files “group1.txt”, “group2.txt”, and “group3.txt”, and the atomic coordinates (extracted from the PDB entry 1JB0 [12]) and respective pigment indices are given in file “trimer.pqr”. These files are available via the supporting online information.

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
