# Peer review of "A Simple Expression for the Screening of Excitonic Couplings between Chlorophylls as Inferred for Photosystem I Trimers"

_ijms, 2024, doi:10.3390/ijms25169006_

Round 1

Reviewer 1 Report

Comments and Suggestions for Authors

This manuscript presents a very nice and informative study of the geometry-dependence of electrostatic coupling in chlorophyll systems. Most importantly, the authors identify two distinct geometric regimes where an exponential screening correction either is or is not necessary to match the more rigorous Poisson-TrEsp method. The simplicity of the final (somewhat empirical) result will make it very useful to the considerable body of researchers interested in understanding intermolecular electronic coupling in photosynthetic systems. I recommend the manuscript be accepted without further review, though I have a few suggestions for the authors that I think would increase impact and readability. 

There are only two substantive points that feel a little unclear to me: Neither is essential for publication, but if the authors can shed some light on them, I think it would substantially boost the impact of the paper: 

1. It wasn't clear to me in the final analysis whether or not there is a simple (known) physical explanation for this geometric dependence to the screening factor. The discussion of various cavity models sheds some light on the values of the prefactors, but (unless I missed it) did not illuminate the reasons for the geometry dependence. This isn't necessary for publication, but it would be very helpful if the authors can either (a) offer an intuitive explanation for the geometry dependence or (b) state explicitly that this isn't yet clear.     

2. How accurate an approximation is the use of a step function in switching from group 2 to group 3 behavior? It feels a little disconcerting physically to have an abrupt jump from one regime to the other as a function of dipole orientation. Is it possible to probe the "width" of this transition through model calculations where the two molecular positions are fixed but one of the dipole moments is progressively rotated? (Similar to Figs 7 and 8 but as a function of angle instead of position.) 

Minor points: (Solely about presentation, not science.)

* The Theory section gives a very nice intro to the Poisson-TrEsp method, but I spent some time puzzled as to how this relates to the "2-cavity" vs "many-cavity" discussion immediately preceding it in the Introduction. It would be helpful to explicitly return to this point at the end of the Theory section, where cavity dielectrics are discussed.

* Closely related: from the Introduction, I was expecting much more discussion of the two-cavity vs. many-cavity question. If I understood correctly, after Figure 3, all results are for the two-cavity model? If so, it would be helpful to state this (and briefly explain why) both at the end of the introduction and in the conclusions. 

* Table 1: I got confused at first (in a black-and-white printout) due to the close spacing of the head-to-tail dipoles at the bottom of the table and the grid lines separating the rows. A little more vertical white space between rows might be helpful. 

* Figure 5: On first glance, I mistook the x-axis (K0) for K itself, and thought the variances were showing that the model did a terrible job for certain relative orientations. It didn't take long for me to realize the mistake, but I wonder if there's a simple way to add a line showing, e.g., the data from the last column of Table 3 in the figure, to highlight that the final model performs well for all relative orientations?  

* Frame labels (e.g., "many-cavity" and "two-cavity") would be helpful in Figure 3. 

Author Response

Comment 1: This manuscript presents a very nice and informative study of the geometry-dependence of electrostatic coupling in chlorophyll systems. Most importantly, the authors identify two distinct geometric regimes where an exponential screening correction either is or is not necessary to match the more rigorous Poisson-TrEsp method. The simplicity of the final (somewhat empirical) result will make it very useful to the considerable body of researchers interested in understanding intermolecular electronic coupling in photosynthetic systems. I recommend the manuscript be accepted without further review, though I have a few suggestions for the authors that I think would increase impact and readability. 

Response 1: Many thanks.

Comment 2:  There are only two substantive points that feel a little unclear to me: Neither is essential for publication, but if the authors can shed some light on them, I think it would substantially boost the impact of the paper: 

1) It wasn't clear to me in the final analysis whether or not there is a simple (known) physical explanation for this geometric dependence to the screening factor. The discussion of various cavity models sheds some light on the values of the prefactors, but (unless I missed it) did not illuminate the reasons for the geometry dependence. This isn't necessary for publication, but it would be very helpful if the authors can either (a) offer an intuitive explanation for the geometry dependence or (b) state explicitly that this isn't yet clear.    

Response 2: Many thanks for pointing out this open point. We agree that the analytical models provide the prefactor, but an explanation of the different distance-dependencies of the two geometries was missing. Some qualitative information can be obtained from the work of Hsu et al. [33] considering two point dipole in a common cavity, as described now on page 4, lines 106-113:

"This effect was qualitatively explained by image dipoles, defined such as to represent the polarization
of the dielectric. The polarization-mediated excitonic coupling then results from the
interaction between the transition dipole of one pigment with the image dipole of the
other. Whereas the image dipole of a dipole pointing at right angle onto a dielectric
surface is oriented parallel to the original dipole, that of a dipole parallel to the dielec-
tric surface is oriented antiparallel thereby enhancing and decreasing, respectively, the
dipole-dipole coupling obtained without the dielectric [33]."

and in the Conclusions section on page 16, lines 396-405:

"For large interpigment distances the screening factor becomes in-
dependent on molecular distance and for in-line and sandwich type geometries adopts
a constant value of 0.6. This value can be qualitatively understood by analytical models
considering point dipoles in spherical cavities. We have to note that the quantitative
agreement of one such models relies on a fortuitous error compensation. At short inter-
molecular distances, where the molecular cavities start to overlap, different behavior is
obtained for in-line and sandwich dimers. Qualitatively, a transition occurs towards a
scenario discussed by Hsu et al. [33] for two point dipoles in a single spherical cavity in
terms of image dipoles causing an enhanced and decreased coupling between in-line
and between sandwich dipoles, respectively (see introduction)."

Comment  3: How accurate an approximation is the use of a step function in switching from group 2 to group 3 behavior? It feels a little disconcerting physically to have an abrupt jump from one regime to the other as a function of dipole orientation. Is it possible to probe the "width" of this transition through model calculations where the two molecular positions are fixed but one of the dipole moments is progressively rotated? (Similar to Figs 7 and 8 but as a function of angle instead of position.) 

Response 3: Many thanks for pointing out this important question. We have noted this question on page 9, lines 254 and 257:

"We note that the use of a Heaviside function in eq 26 is a simplification, since the screening factors calculated with Poisson-TrEsp undergo a more smooth transition with respect to the mutual orientation of pigments, as will be discussed in more detail below with model calculations."

and we have rotated one of the pigments in the in-line and sandwich model dimers and investigated the screening factors, as described  on pages 13/15, lines 328-342:

"In order to investigate the change of the screening factor with varying orientation for fixed interpigment distance (in this case 12 Å), we have rotated one of the two pigments in the in-line and sandwich dimers around an axis perpendicular to their transition dipole moment (Figure 7). As seen in the lower left panel in Figure 8, for the sandwich dimer the screening factor practically stays constant close to 0.6, in agreement with the empirical screening function f mn (eq 26) since | κ mn | < κ 0 for all rotation angles Ï• of the sandwich dimer (SI, Figure S2). For the in-line dimer, we have | κ mn | < κ 0 for | Ï• | > 54 â—¦ (SI, Figure S2), and the screening factor is roughly constant close to =0.6 (The changes for very large Ï• are not critical, since the couplings are small for these angles. In contrast, for | Ï• | < 54 â—¦ the screening factor steadily increases until it reaches a maximum value of 0.8 at Ï• = 0. Our empirical screening function f mn (eq 26) at | Ï• | = 54° makes a jump but stays constant for smaller angle magnituds | Ï• | < 54 â—¦ , as shown by the black-dashed line in the lower left panel of Figure 8. Hence there is a quantitative
difference between Poisson-TrEsp and the empirical screening function, but the qualitative behavior goes in the same direction."

Figure 7 has been modified to illustrate this rotation and the lower left panel in Figure 8 was added to present the resulting screening factors.

Comment 4:  Minor points: (Solely about presentation, not science.) The Theory section gives a very nice intro to the Poisson-TrEsp method, but I spent some time puzzled as to how this relates to the "2-cavity" vs "many-cavity" discussion immediately preceding it in the Introduction. It would be helpful to explicitly return to this point at the end of the Theory section, where cavity dielectrics are discussed

Response 4: We agree that a more balanced description of the two models is appropriate.  We added the following statement in the end of the theory section (on page 7, lines 190-194: "Please note that in
our treatment we distinguish between the two-cavity and the many-cavity model, as described in the introduction. Whereas in the two-cavity model there are only cavities for those two pigments for which the coupling is determined, in the many-cavity model also the cavities of the remaining pigments are included (Figure 2)."

Comment 5: Closely related: from the Introduction, I was expecting much more discussion of the two-cavity vs. many-cavity question. If I understood correctly, after Figure 3, all results are for the two-cavity model? If so, it would be helpful to state this (and briefly explain why) both at the end of the introduction and in the conclusions. 

Response 5: We have included  the results of the many-cavity model in Fig. 4 and the variances with respect to results obtained with an empirical screening function in Table S1 in the supporting information, as described on page 8, lines 239-241: "The larger scattering of data points obtsined with the many-cavity model in Figure 3 translates into a larger scattering of the data points of the individual groups in Figure 4, but the same fit functions can be used for, both, the two-cavity and the many-cavity
model." ,  on page  10, lines 266-267: "The variances obtained in the many-cavity model are somewhat larger but show the same qualitative behavior (SI, Table S1)." and in the Conclusions section on page 16, lines 410-414: "Finally, we note that the results obtained in the two- and in the many-cavity models are qualitatively very similar. In particular, the same empirical screening function is obtained in the two models. In general, the orientation and distance dependence of the screening factors in the many-
cavity models is less smooth, because of the less homogeneous dielectric environment."

Comment 6: Table 1: I got confused at first (in a black-and-white printout) due to the close spacing of the head-to-tail dipoles at the bottom of the table and the grid lines separating the rows. A little more vertical white space between rows might be helpful. 

Response 6: Thank you, done as suggested.

Comment 7: Figure 5: On first glance, I mistook the x-axis (K0) for K itself, and thought the variances were showing that the model did a terrible job for certain relative orientations. It didn't take long for me to realize the mistake, but I wonder if there's a simple way to add a line showing, e.g., the data from the last column of Table 3 in the figure, to highlight that the final model performs well for all relative orientations?  

Response 7: We modified Figure 5 by including horizontal blue-dashed lines that provide the variances of the different empirical fit functions.

Comment 8: Frame labels (e.g., "many-cavity" and "two-cavity") would be helpful in Figure 3. 

Response: Thanks, done as suggested.

Reviewer 2 Report

Comments and Suggestions for Authors

Authors established the screening method for quantitative excitonic coupling particularly on pigment molecules in photosynthetic apparatus using a chemical polarizable continuum model. No fatal errors were found, thus the present manuscript would be suitable for publication after considering the following point. The final goal of this manuscript is to show the screening method. However, examples found in structures might help to support/understand the results and conclusions for readers.

Author Response

Comment: Authors established the screening method for quantitative excitonic coupling particularly on pigment molecules in photosynthetic apparatus using a chemical polarizable continuum model. No fatal errors were found, thus the present manuscript would be suitable for publication after considering the following point. The final goal of this manuscript is to show the screening method. However, examples found in structures might help to support/understand the results and conclusions for readers.

Response: Thanks, we have included two examples of pigment pairs in the supporting material (Figure S1) which were used in the past to infer an exponential distance dependence of the screening factor and mention these examples in the Discussion section on pages 10/11, lines  279-283: "Two prominent
examples of this study, the P D1 -P D2 pair of the reaction center of photosystem II and the Chl 602 -Chl 607 dimer of the LHC-II light-harvesting antenna of plants were investigated with Poisson-TrEsp and show the expected exponential distance dependence of the screening factor (SI, Figure S1)."